# Neurotrauma clinicians' perspectives on the contextual challenges associated with traumatic brain injury follow up in low-income and middle-income countries: A reflexive thematic analysis

Brandon G. Smith[1,2]ʘ*, Charlotte J. Whiffin[2,3]ʘ, Ignatius N. Esene[4], Claire Karekezi[5], Tom Bashford[2], Muhammad Mukhtar Khan[2,6], Davi J. Fontoura Solla[2,7], Bhagavatula Indira Devi[2,8], Wellingson S. Paiva[2,7], Franco Servadei[9], Peter J. Hutchinson[1,2], Angelos G. Kolias[1,2]*, Anthony Figaji[2,10], Andres M. Rubiano[2,11]

1 Division of Neurosurgery, Department of Clinical Neurosciences, Addenbrooke's Hospital and University of Cambridge, Cambridge, United Kingdom, 2 NIHR Global Health Research Group on Neurotrauma, University of Cambridge, Cambridge, United Kingdom, 3 College of Health, Psychology and Social Care, University of Derby, Derby, United Kingdom, 4 Neurosurgery Division, Faculty of Health Sciences, University of Bamenda, Bambili, Cameroon, 5 Neurosurgery Unit, Department of Surgery, Rwanda Military Hospital, Kigali, Rwanda, 6 Northwest School of Medicine & Northwest General Hospital & Research Centre, Peshawar, Pakistan, 7 Division of Neurosurgery, Department of Neurology, University of São Paulo, São Paulo, Brazil, 8 Department of Neurosurgery, NIMHANS, Bangalore, India, 9 Humanitas Research Hospital-IRCCS and Humanitas University, Rozzano, Milan, Italy, 10 Division of Neurosurgery, Red Cross Children's Hospital & University of Cape Town, Cape Town, South Africa, 11 Neurosciences Institute, El Bosque University, Bogotá, Colombia

ʘ These authors contributed equally to this work.
* bgs30@cam.ac.uk (BGS); ak721@cam.ac.uk (AGK)

## Abstract

### Background

Traumatic brain injury (TBI) is a major global health issue, but low- and middle-income countries (LMICs) face the greatest burden. Significant differences in neurotrauma outcomes are recognised between LMICs and high-income countries. However, outcome data is not consistently nor reliably recorded in either setting, thus the true burden of TBI cannot be accurately quantified.

### Objective

To explore the specific contextual challenges of, and possible solutions to improve, long-term follow-up following TBI in low-resource settings.

### Methods

A cross-sectional, pragmatic qualitative study, that considered knowledge subjective and reality multiple (i.e. situated within the naturalistic paradigm). Data collection utilised semi-

ᴼᴾᴱᴺ ACCESS

**Data Availability Statement:** Data cannot be shared publicly because of our ethical agreements that do not allow for raw data to be made widely

available as there would be identifying information, and allowing public access of these would go against participant confidentiality. Additionally, there may be potentially sensitive or controversial information of which may be at risk of being misconstrued if made publicly available. Anonymized transcripts are available from the National Institute for Health Research (NIHR) Global Health Research Group on Neurotrauma (https://neurotrauma.world/) for researchers who meet the criteria for access to confidential data, or by e-mailing Carole Turner (group project coordinator, clt29@medschl.cam.ac.uk). The authors did not receive any special privileges in collecting or accessing the data that other researchers would not have.

**Funding:** This research was supported by the National Institute for Health Research (NIHR) Global Health Research Group on Neurotrauma (grant number 16/137/105) using UK aid from the UK government. The funders had no role in study design, data collection and analysis, decision to publish, or preparation of the manuscript.

**Competing interests:** AGK and PJAH are supported by the National Institute for Health Research (NIHR) Cambridge Biomedical Research Centre, Royal College of Surgeons of England, and the NIHR Global Health Research Group on Neurotrauma. PJH is also supported by an NIHR Research Professorship. The NIHR Global Health Research Group on Neurotrauma was commissioned by the UK NIHR using Official Development Assistance funding (project no. 16/137/105). INE, CK, MMK, DJFS and AGK are members of the Young Neurosurgeons Committee of the World Federation of Neurosurgical Societies. The committee is supporting this project.

structured interviews, by videoconference and asynchronous e-mail. Data were analysed using Braun and Clarke's six-stage Reflexive Thematic Analysis.

## Results

18 neurosurgeons from 13 countries participated in this study, and data analysis gave rise to five themes: Clinical Context: What must we understand?; Perspectives and Definitions: What are we talking about?; Ownership and Beneficiaries: Why do we do it?; Lost to Follow-up: Who misses out and why?; Processes and Procedures: What do we do, or what might we do?

## Conclusion

The collection of long-term outcome data plays an imperative role in reducing the global burden of neurotrauma. Therefore, this was an exploratory study that examined the contextual challenges associated with long-term follow-up in LMICs. Where technology can contribute to improved neurotrauma surveillance and remote assessment, these must be implemented in a manner that improves patient outcomes, reduces clinical burden on physicians, and does not surpass the comprehension, capabilities, or financial means of the end user. Future research is recommended to investigate patient and family perspectives, the impact on clinical care teams, and the full economic implications of new technologies for follow-up.

## Introduction

Traumatic Brain Injury (TBI) is a major global health issue; but low- and middle-income countries (LMICs) face the greatest burden [1, 2]; an estimated 69 million cases of TBI per year are reported worldwide, with LMICs facing almost three times as many cases than high-income countries (HICs) [3]. Although death from neurotrauma is decreasing globally, TBI causes significant disability from cognitive, emotional and/or physical sequalae [4–7]. However, the true burden of TBI is unknown [8] because despite improvements in the collection of data, morbidity and long-term functional outcomes remain inadequately recorded in both LMICs and HICs [8–11]; for example, many of those who die from TBI in LMICs are not taken to local hospitals, and therefore not recorded in mortality rates. Therefore, while neurotrauma outcomes in LMICs may differ from those in HICs, the real magnitude of this difference cannot be accurately quantified [12].

In 2004 the World Health Organisation (WHO) released Guidelines for Essential Trauma Care, bringing to the fore the role of surveillance and outcome data in reducing the global burden of mortality and morbidity as a result of injury [13]. This data enables clinical teams to determine the degree and breadth of physical, mental, cognitive and socio-economic sequalae post-injury [9, 11, 14, 15]. Accurate data also facilitates the evaluation of systems and services, enabling: identification targets of wider systems improvement in injury prevention [16], assessment of the efficacy of management decisions and patient treatment [11, 15], facilitation of quality improvement initiatives and clinical trials, and lastly, the establishment and continuation of registries. These activities may serve to facilitate the development of care pathways, health policy, and injury prevention strategies [17–19].

Outcome data is therefore an important part of improving global TBI clinical care, which was the main strategic goal of the National Institute for Health Research (NIHR) Global Health

Research Group on Neurotrauma (GHRGN) established in August 2017 in Cambridge, U.K. This group initially collaborated with twelve LMIC partners (see Box 1) and identified a need to improve the collection of long-term outcomes following TBI in LMICs. However, given the heterogeneity within and between LMICs, the process of outcome data collection is considered more complex [3, 11, 20], and is often restricted to the point of hospital discharge [11]. Why long-term outcomes are not consistently recorded has not been empirically examined [21], and equally, very little has been published on the obstacles faced in delivering comprehensive, timely long-term follow-up in LMICs. Of the limited literature, a lack of regular follow-up for trauma patients was perceived to be secondary to circumstances such as weak healthcare, lack of centralised digital records, and long-term support infrastructure [11, 22]. Technologies such as SMS, telephone and smartphones have been shown as a feasible and cost-effective option for follow-up in both neurosurgical populations and other clinical fields, and may begin to alleviate the challenges of distance and expense of travel in service access [22–25].

---

### Box 1. Collaborating low- and middle-income countries in the NIHR GHRGN

- Brazil
- Colombia
- Ethiopia
- India
- Indonesia
- Malaysia
- Myanmar
- Nigeria
- Pakistan
- South Africa
- Tanzania
- Zambia

---

Therefore, the aim of this study was to explore the specific contextual challenges of, and possible solutions to improve, long-term follow-up and outcome data collection following TBI in countries identified as low- or middle-income as defined by the World Bank [26, 27].

## Methods

### Study design

The Consolidated criteria for Reporting Qualitative Studies (COREQ) guidelines were used to report this study [28].

This was a cross-sectional, pragmatic, qualitative study, situated within a naturalistic constructivist paradigm. Naturalistic inquiry tries to stay true to the nature of the phenomena under investigation by staying close to the natural social world of the participant [29]. Findings are regarded as a creation of an interdependent interaction between inquirer and participant with the researcher serving as the instrument for data collection, grounding and situating their findings in the data collected [30].

Pragmatic qualitative research is not aligned to a specific theoretical perspective. As such methods are more flexible and allow the study to be designed in way that best suits the research question [31]. The aim of inquiry is to reach an in-depth understanding prioritising description first. Deeper analysis follows through interpretation of how people draw meaning from their experiences [32]. Conducting the study in this way allows for interpretation and realisation of individuals roles in real-world situations, identifying key phenomena and understanding these through their contextual backdrop [33]; in turn, qualitative methodology offers researchers the means to conduct an in-depth, immersive inquiry, fostering a rich understanding of the human experience [34].

## Ethics

The University of Cambridge Psychology Research Ethics Committee reviewed this study (PRE.2020.010). Participants provided informed written electronic consent through Qualtrics[TM] (Qualtrics, Provo, UT, USA). Participants were able to withdraw from the study at any time and their contribution was confidential. All data were anonymised as far as possible, and participants were cautioned that full anonymity might not be possible when direct quotes are presented in manuscripts. Participants were given the option to be a named collaborator in recognition of their contribution and cautioned that this might affect their anonymity. We recognise that incentives are controversial in research, and yet reasonable remuneration for time and inconvenience are common. Therefore, without the ability to financially remunerate participants, we felt being a named collaborator reflected their commitment to this project in a population who were extremely limited in their available time.

## Sampling & recruitment

Purposive sampling was used in this study, in which participants are selected who are 'information-rich' because they are knowledgeable in the phenomenon of interest (Table 1) [35]. In qualitative research sample size is often informed by principles of data saturation. However, determining sample size *a priori* is seen as 'inherently problematic' [36], and principles of saturation no longer the primary mechanism for determining sampling sufficiency in reflexive thematic analysis [37]. Therefore, at the start of the study we made a pragmatic decision to estimate sample size based on having participants from each of the 12 collaborating countries in NIHR GHRGN in addition to new group collaborators from the Philippines and Zimbabwe.

**Table 1. Inclusion criteria.**

| Inclusion criteria |
| --- |
| • Practising physician within a country identified as an LMIC by the World Bank |
| • At least 2 years' experience of managing neurotrauma |
| • Some experience in the research process and/or long-term follow-up |
| • Self-declared fluency in written or spoken English |
| • Access to electronic communication platform (e-mail, videoconferencing, telephone) |
| • Able to provide informed consent |

We were also mindful of the resources available, our methodological approach and our analytical plan. We therefore estimated that 24–48 participants would be feasible and sufficient to address the aims of the study. While in quantitative studies these sample sizes may be thought of as small, such samples are more methodologically appropriate in qualitative studies and facilitate rich, in-depth analysis and understanding of the data.

The first call for participants was by email from to the mailing list for the group in March 2019 by the project co-ordinator. Second and third calls were made two and three weeks later. Unfortunately, we were unable to reach our minimum target size using this method, and therefore we amended the protocol to allow us to recruit via social media (specifically Twitter and WhatsApp). This recruitment method was compliant with the terms and conditions of both Twitter and WhatsApp An infographic advertising recruitment to the study was re-tweeted several times by the author team, and members of the NIHR GHRGN, to increase the likelihood of reaching the required sample size. Potential participants expressing interest through social media were invited for further communication by e-mail.

Those who responded to the invitation were sent a participant information sheet and invited to a pre-consent meeting to answer questions where eligibility was assessed.

A full list of the inclusion criteria applied can be found in Table 1. Where the inclusion criteria were not met, interested parties were deemed ineligible for further participation.

## Data collection

This study used in-depth semi-structured interviews. Semi-structured interviews are an effective way to collect open-ended data, and are designed to garner subjective responses from the participants regarding their experiences, perspectives or phenomena they have experienced [38]. An interview schedule is particularly useful for when the researcher possesses an ample degree of objective background knowledge regarding the experiences and phenomena faced by the participants, yet does not possess the subjective knowledge themselves [38]. This interview style offers many benefits versus their formal, structured counterpart. Semi-structured interviews allow for a greater degree of exploration and probing of the world of the participant, facilitates rapport and empathy as conversations are merely guided by the interview schedule rather than strictly dictated in a linear question-response manner. Furthermore, they enable the researcher to follow the participant's narrative and explore novel, deeper lines of enquiry previously undefined by the schedule, yet remaining within the scope of the research questions posed [39]. All interviews were held on-line through a videoconferencing application and conducted by the lead author, an MBPhD student with formal qualitative methods training. The interview guide (see Box 2) was constructed in consultation with co-authors to ensure these

---

### Box 2. Semi-structured interview guide

#### Current follow-up

A  Please can you tell me about how and when your patients are currently followed-up following discharge from a TBI?

　a  What outcomes are measured?

　b  What needs are assessed?

---

 c What referrals are made?

 d What type of contact system do you have in place (face-to-face, telephone, telemedicine)?

## Definition and understanding of long-term outcomes

A What do you understand by the use of 'long- term' in the context of outcomes post-TBI?

B Do you think your definition and understanding of 'long-term outcomes' in your country are the same as those in other countries?

C Do you know or use any type(s) of outcome classification for follow-up?

## Attitudes towards long-term follow-up

A What do you think about the need for long-term follow-up of patients following TBI in your setting?

B What would you consider to be the main benefits of long-term follow-up of TBI patients?

C What do you consider to be the challenges associated with long-term follow-up of your patients with TBI?

D Are these challenges more related to: health system aspects, national, regional, or local administrative aspects, institutional aspects, resources for care aspects?

E Are there any other issues that need to be considered for long-term follow-up in your own setting and wider country?

## Long-term follow-up in LMIC

A Thinking more broadly now about other Low- and Middle-Income Countries, do you have any other thoughts about long-term follow-up in these other LMICs

B What else might be required in LMICs to facilitate long-term follow-up of patients following TBI?

## Possible interventions to facilitate follow-up in LMIC

A Do you know what types of technology patients and families have access to that could help in follow-up of patients following TBI?

B What kind of technology would help in your country to record long-term follow-up?

C Are you aware of any specific mobile health, telemedicine, or phone interview options for long-term follow-up in your institution/state/country?

D Does your service or institution participate in a trauma or neurotrauma quality improvement program including the use of clinical registries with outcome measures?

were appropriate and were available to participants in the participant information sheet given to them prior to consent.

No issues with language, nor arranging consent and interview sessions across multiple time zones, were experienced. In a minority of cases, connection and videoconference quality was temporarily impeded but was quickly remedied by the participant moving location to achieve a better network connection, or both participant and interviewer turning off their webcam for the remainder of the session to resolve internet bandwidth issues. Interviews lasted on average 38 minutes (range 27–55 minutes).

Although in-person interviews are the cornerstone of qualitative methods and often the preferred method of data collection [40, 41], videoconferencing has been demonstrated as a viable and alternative medium to conduct qualitative data collection [42–44]. However, two participants who expressed interest in the study were unable to complete an in-depth interview. Both asked if the interview could be conducted via email instead. Being inflexible in the method of data collection risked losing these important insights from the study. Therefore, we amended the study to allow email interviews to be undertaken. In these cases, the questions were sent one at a time to facilitate discussion and the responses from participants were suitably rich and detailed. These methods facilitated international recruitment by overcoming the geographical barriers and offered equitable access to all [42, 43, 45, 46].

Though no separate pilot study was conducted, after three interviews, the interview guide was reviewed between co-authors and critiqued with respect to its administration and content. An in-depth, reflective discussion on the interview style of the lead author was also held at this time. No further revisions were made to the protocol or guide at this stage, and feedback relating to interview style was incorporated into further interviews.

In addition to the qualitative data, we collected a small amount of demographic data to ensure we captured a contextual understanding which would be relevant to subsequent interpretation of the data. We used Qualtrics$^{TM}$ to collect age, sex, country of residence, years of experience, years working in neurotrauma, and practice setting.

## Data analysis

We used Braun and Clarke's 'Reflexive Thematic Analysis' (RTA) [47–50] which is a specific analytical approach requiring six stages to interpret participants perceptions and experiences and represent these as themes (see Table 2).

Audio recordings were sent for translation by a third-party translation service. Upon return, transcripts were checked for accuracy by replaying the audio files and read several times to aid immersion. Independent coding by B.G.S. was supported by NVivo$^{TM}$ 12 (QSR International, Melbourne, Australia). Codes were not identified a priori, and coding proceeded in an inductive and iterative manner. Development of themes was not formulaic; instead, potential similarities and relationships between codes were explored leading to the construction of early themes. Themes were continually reflected on, checked, and refined, ensuring good representation of participant's views. Frequent debriefing sessions were held between lead authors (B.G.S. and C.J.W.) to review understanding and advance interpretation. Findings were shared with both the co-author team and participants. Feedback helped us to further engage with the data and be confident of our final interpretation.

One point of recurring feedback from co-authors was sample sufficiency and data saturation. However, data saturation remains a much debated issue in qualitative research and such claims are said to be better avoided in RTA [37]. A more consistent claim would be that we reached a point where the analysis of new data did not alter the themes we had constructed

**Table 2. Application of reflexive thematic analysis [47, 49, 50].**

| Phase | Description | Product of framework |
|---|---|---|
| **1. Data familiarisation and writing familiarisation notes** | Transcription of data, reading and re-reading data, taking note of initial ideas and keeping a reflexive diary. | Preliminary codes. |
| **2. Systemic data coding** | Systemically coding interesting features of the data across the dataset, collating data relevant to each code, and creating relevant memos as to the meaning behind each code. | Comprehensive coding. |
| **3. Generating initial themes from coded and collated data** | Collating codes into potential early categories and exploring hierarchies and possible relationships between these, with critical discussion with second author. | Raw, potential 'precursor themes'. |
| **4. Developing and reviewing themes** | Checking if the themes work with relation to the coded extracts, and further with the entire dataset, followed by generating a thematic map of the analysis, with further critical discussion with second author. | Themes and an early map of the potential relationships and flow between themes. |
| **5. Refining, defining and naming themes** | Ongoing analysis and revisiting and refining specifics of each theme, the overall narrative being portrayed from the analysis, and generating firm names and definitions for each theme with second author. | Refined themes and a uniting narrative. |
| **6. Writing the report** | Selection of vivid, compelling extract examples and a final opportunity for analysis, relating the analysis to the initial research question and literature in the production of a first report. | Comprehensive report of all themes, interpretations, and accompanying narrative supported by quotes. |
| | Respondent validation and co-author feedback advanced and finalised the final report. | |

and by the end of analysis we were confident we had generated sufficient insight and understanding to answer the research question.

## Rigour

In this study we used a number of methods to enhance the quality of the findings. These included: Ensuring methods were congruent with the underpinning philosophy of the study; full immersion in the analytical process through repeat engagement with raw data; close supervision and reflexive conversations during data analysis; maintaining a reflexive journal to document emerging understanding of the data; and challenging naïve interpretation through dialogue with co-authors and participants. The international research team were, demographically, ethnically, culturally, and professionally diverse. The co-author leads are White British; B.G.S. is an 7th year MBPhD student with formal qualitative training. C.J.W. is a Senior Lecturer in Nursing and experienced qualitative researcher. T.B., I.N.E., C.K., M.M.K., D.J.F.S., B.I.D., W.S.P., F.S., P.J.H., A.G.K., A.F., A.M.R. are all physicians or neurosurgeons with an extensive range of experience in neurotrauma. Finally careful use of NVivo and its associated functionality such as memos and annotations enabled a clear audit trail to be created from raw data to final themes. This strong commitment to reflexivity and critical discussion throughout the study facilitates transparency and improves credibility in qualitative research [51, 52].

## Inclusivity in global research

Additional information regarding the ethical, cultural, and scientific considerations specific to inclusivity in global research is included in S1 Checklist.

## Results

During April 2020 –March 2021, we received 55 expressions of interest from neurosurgeons in LMICs. Nineteen potential participants were consented; however, one participant was deemed to withdraw by default after loss of contact prior to interview. The remaining 36 individuals did not respond after information sheets were supplied—our reflections on this were that

many of these expressions of interest were from those who wanted to conduct the study rather than be a participant in the research. Furthermore, we commenced recruitment at the height of the pandemic and many neurosurgeons did not have the time to commit to an in-depth interview. At the end of recruitment 18 neurosurgeons participated in this study (n = 16 semi-structured interviews, n = 2 email interviews) (Table 3).

Five themes were identified from the data: 1) Clinical Context: What must we understand? 2) Perspectives and Definitions: What are we talking about? 3) Ownership and Beneficiaries: Why do we do it? 4) Lost to Follow-up: Who misses out and why? 5) Processes and Procedures: What do we do, or what might we do? these are now discussed through their related subthemes.

## Clinical context: What must we understand?

Theme one provides an important description of LMIC neurotrauma practice and TBI care and provides a contextual backdrop to all subsequent findings. Subthemes were: context of care; patient pathway; and healthcare economics.

**Context of care.** Participants described similar challenges but different capacities to manage them. Differences in neurotrauma provision created disparity in access from centralisation of specialist facilities which necessitated lengthy travel for those in more rural areas. High disease burden created huge demand, and concerns were raised that they did not have enough time. International comparisons were often drawn and neurotrauma in LMICs was perceived

**Table 3. Participant demographics.**

| Age | | World Bank Income Classification | |
|---|---|---|---|
| 30–39 | 11 | Low | 4 |
| 40–49 | 2 | Lower-middle | 10 |
| 50–59 | 5 | Middle | 4 |
| **Sex** | | **World Bank Regional Unit** | |
| Male | 15 | East Asia & Pacific | 1 |
| Female | 3 | Europe & Central Asia | 0 |
| **Practice setting** | | Latin America & the Caribbean | 2 |
| Public | 15 | Middle East & North Africa | 2 |
| Private | 8 | South Asia | 6 |
| Urban | 9 | Sub-Saharan Africa | 7 |
| Rural | 1 | **Country of residence** | |
| Large national reference centre | 9 | Brazil | 1 |
| Small regional centre | 3 | Colombia | 1 |
| **Years of experience post-registration** | | Egypt (Arab Rep.) | 1 |
| 2 to 5 | 9 | Ethiopia | 2 |
| 6 to 10 | 2 | India | 3 |
| 11 to 15 | 2 | Malaysia | 1 |
| 15 + | 5 | Morocco | 1 |
| **Years of experience in neurotrauma** | | Nepal | 1 |
| 2 to 5 | 5 | Nigeria | 2 |
| 6 to 10 | 5 | Pakistan | 2 |
| 11 to 15 | 1 | Rwanda | 1 |
| 15 + | 7 | South Africa | 1 |
| | | Uganda | 1 |

as worse, with a higher incidence per capita, and a wider spectrum of disease pathologies, than HICs.

*"[. . .] there is a lot of patients. And there are too many things you need to give to each patient, to give attention, but you can't. You can't. Exactly you can't. Because in one day outpatient follow up is 120, 150 patients. So it is very difficult to give each patient sufficient time. So you just look at the- if the patient was discharged for a short time, since a few weeks, you look at the wound, you remove the sutures, and you check if there is any problem with the medication"* [G]

*"Just one CT scanner for, for the city, for one city. For city or for maybe I don't know two or three million inhabitants. So it is very difficult here."* [M]

*"In my State we have neurosurgery services in the main city, the capital. In one, just one more city that, that is 250 kilometres from here. So from the place, from a city in the south of the state to the city we have 5,000 kilometres with no neurosurgical service"* [J]

Despite these contextual challenges, many 'pockets of excellence' were also reported that were considered on par with, or exceeding, those in HICs.

*"Now some of those states, [. . .] they have an incredible programme of rehab. And the kind of rehab that would dwarf the rehab programmes that I've seen in the UK and the US. Because they have clearly been driving it for a long time. [. . .] So, there are these little pockets of excellence. But the spectrum of course is a very wide one. And it's a graded spectrum. So, you will see everything from absolutely no follow-up at all, where it's all about survival, to situations where the follow-up and the rehab services is so good it's on par with the best that one could see in the developed world circumstances. And then everything in between"* [O]

**Patient pathway.**   Key milestones were described within the patient pathway and patients were understood to have numerous ongoing physical, psychological, and cognitive sequelae that warranted further investigation. However, there were obstacles in providing care, such as a sparsity and centralisation of both specialist and rehabilitative services, inequitable access to services, weak organisation of existing infrastructure, and reduced clinician availability owing to case volumes. A stark contrast was made between paediatric and adult services, whereby adult care was considered sporadic and less comprehensive.

*"There's a secondary injury that we try to prevent. There's also the same secondary injury that leads to chronic morbidity. But of course, you can't run a severe TBI programme without incurring morbidity in survivors. So that's a given. And most surgical departments are very focused on the acute care, but less so focused on the chronic care. But of course, we know because of the morbidity that's as important as the acute care. So, patients are always going to have need of occupational therapy, physiotherapy, speech therapy, neuropsychological, neuro-psychiatric support."* [O]

*"It differs as the injured brain is quite heterogeneous even with same severity, different patient demography and genetics, etc."* [C]

*"the patient and follow-up requires multi-speciality. Not only neurosurgery, he requires physiotherapy some of the eye problems they thought he might be requiring a visit to a ophthalmology centre. Or some ear problem so he might be requiring a visit to an ENT specialist [. . .]it's not feasible in a day [. . .] this is one of the most important difficulties which we face"* [P]

*"We should not consider traumatic brain injury as an event but as a disease process."* [H]

*"The [adult] follow up is a lot more sporadic. So, they will get- they are not followed up in the long run in the neurosurgical outpatients like we do in children. But patients with particular problems will be referred to various specialties. So, whether that be rehab services like occupational therapy, physiotherapy, or neuropsychiatric, or neuropsychological follow up. So, those clinics also exist. But those clinics are generally in adult services quite overwhelmed. So, it's only really patients who have major, major issues. But the follow up isn't as comprehensive in adult services at all."* [O]

**Healthcare economics.**   Financial burden of TBI was described both in terms of direct care costs and lost employment. Despite efforts to provide charitable and state-funded care, economic inequality remained. Where private practices and health insurance schemes existed, these were often out of reach for patients. Demand for care from new and returning patients also burdened healthcare resources. Lack of clinicians, imaging, clinics, and beds were described as causes of reduced capacity that could pose hard limits on treatment beyond acute care. A 'survival-first' model of care was described by many, with in-depth examinations and investigations considered a 'luxury'.

*"The state hospitals, like I told you, have such huge numbers that they couldn't even begin to think of follow-up. They treat and they send. [. . .] [here] they have this huge number of patients who come in. They do very good work. But the moment they finish the surgery, place you on a ventilator, they send the patient away into the hospital they came from. [. . .] So, they have no idea what happens after that. They do the immediate acute care. And then they run"* [F].

*"The most important reason in the finance is the cost. [. . .] The cost is the most important thing. Lack of health insurance and the patient has to pay from his own pocket."* [E]

*"I ask them, why didn't you come earlier? [They said] 'I don't have money to come here.'"* [G]

*"Clinicians or service providers end up focusing on the bare minimum. So, it ends up being all about survival."* [O]

Stark ratios between neurosurgeons and the population were described and this overburdening wore down morale and exacerbated gaps in provision.

*"And then because they are overburdened in terms of their work rate it's easy for them to get into this vicious spiral of caring less and less [. . .] it's just the constant burden of disease that kind of wears them down. So, fatigue sets in. Cynicism sets in. All of that. And so, you already have reduced resources and you burn out the people who are then trying to provide those resources even more. [. . .] Fewer people providing the services so they get dumped on and have reduced capacity, psychologically, emotionally, physically, financially."* [O]

The importance of both intra- and inter-hospital coordination, and shared responsibilities in multi-disciplinary follow-up, was highlighted.

## Perspectives and definitions: What are we talking about?

Theme two describes participant views on what constitutes 'long-term' follow-up, and capacity to offer follow-up within these parameters and included two sub-themes: Duration and intervals and capacity and practice.

**Durations and intervals.** Several definitions of 'long-term' were identified, varying from the point of discharge to at least one, three- and six-months post-discharge, to time spans of six to 12 months, one year, two years, five years and lifelong. However, the most frequently reported definition was six to 12 months in adult populations.

> *"I think once we've discharged the patient from our hospital, okay, long-term can be anything just from a first follow-up to the last follow-up. [. . .] long-term can be at least for three months. [. . .]can be three months to any number of years. The longer it is the better it is. [. . .] The long-term is definitely needed, the longer it is the better you know about the disease." [P]*

> *"So, I think unfortunately not still many people in the community of the neuro trauma care have the real concept of long-term outcomes, many people just think there is just at the end of the hospitalisation plan discharge or sometimes just one and between six months after discharge." [I]*

In contrast, long-term in a paediatric setting was described as at least four to five years and was explained as necessary to mitigate the possibility of mis-assessment of the child's level of functioning, and to distinguish between cognitive or behavioural sequelae versus usual childhood behaviour. With respect to the intervals between outpatient appointments following discharge, there were clear similarities between settings, and often offered at one-, three- and six-month intervals, followed by annual visits.

**Capacity and practice.** Despite some similarity in the timing of follow-up, many also described follow-up as 'needs-driven', with complex cases necessitating more tailored and extensive review. Needs-driven follow-up was also in response to the demand for care and a necessity for prompt discharge.

> *"Um, I think it varies between places [. . .] I think it depends on the setup and in the patient load of the hospitals and so on [. . .] we have clinics with many patients so if they are having like no new complaints and so on we tend to like reassure them and discharge them after six months but yeah I think it is built on the based on the settings that we have in developed countries yeah." [A]*

Many participants expressed a desire to follow-up patients in a way that would more closely reflect their theoretical understanding of long-term follow-up, but were constrained in their ability to achieve this.

## Ownership and beneficiaries: Why do we do it?

Theme three was identified through the clinician's portrayals of stakeholders in follow-up: Patient; Physician; State; and Science.

**Patient.** Patients were regarded almost ubiquitously as the primary beneficiary of follow-up. The heterogenous sequelae of TBI, and varying needs post-injury were acknowledged. Multiple longitudinal assessments were believed to be advantageous, providing the best chances of recovery.

> *"I think the long-term follow-up, the most important benefit is we know about the disease fully. How that person reacts. How the person responded. Or how the person's brain responded to the disease fully, in a long-term way. We also know about the rehabilitation, which is required okay, for the particular disease. Okay, in terms of vocational, in terms of every aspect of the injured patient." [P]*

*"to develop a proper compensation mechanism for the TBI patients to cope with the long-term sequelae, and early therapies to delay or to reduce the long-term impact among brain-injured patients by earlier initiation of appropriate management (such as cognitive rehabilitation) or medications with long-term follow-up."* [H]

*"For me, the patient will get a confidence in, in his self and he will understand more about his injury and that is also will help if maybe we put on epileptic drug which will, we should remove that slowly with the check-up of each neurologists. Sometimes we use that for people that maybe are coming with seizure, with epilepsy, post-trauma. That also to help them, to help them to make a follow up on those drugs [. . .] what can I say that I think that the benefits is not just for us it is also for the patient because always patients will, will have contact with his, with his doctor and after refer they have some things to, new happen for him."* [M]

*"I just received a couple of days ago a request from a patient that I operated in 2012 for a severity TBI with a bilateral decompressive craniectomy because he needs a summary a very good summary of the whole process that we have been doing with him with this data collection because he is asking for a permit of disability for economical support from the government [. . .] our summary will be really important. So, we never think about that type of conditions sometimes when you are doing the following of the patients and this is a case that illustrates very well the importance of this kind of follow up."* [I]

Regular follow-up identified complications early and managed sequelae as they arose, or provided the means to refer on for additional therapies, interventions or investigations. Follow-up was also seen by many as an opportunity for patient counselling, education, and both vocational and occupational support, the latter regarded by a few as one of the most important facets of long-term care; assisting the patient's successful reintegration into home, employment, and wider society.

**Physician.**   Physician benefits included a means to inform resource allocation, and aid strategizing/modifying therapies (where treatment modification reduced healthcare costs this was seen as an additional patient benefit.

*"And the other thing is like we can recognise the factors in long term follow up that can be minimised. The treatment protocols like if you are giving one treatment, and if you are doing one long term follow up with the patient, so the patient, we cannot through research and through long term follow up, so we follow the patient for long term so we can do research and learn many things, like treatment protocols, like what treatment is effective, which treatment is ineffective."* [E]

*"with the one monthly follow-up, I'm learning more on who, where to focus our treatment on. Because we cannot treat everybody optimally. We don't have the facilities. Do you understand? [. . .] we need to know who did well, who did badly. And then tailor our care management so that we spend more of the resources on the ones who will do well. And recognise the ones who won't do well early."* [F]

Personal learning, professional development, and the opportunity to enhance the patient-doctor relationship were also perceived as benefits.

**State.**   Successful follow-up and, where possible, reintegration into society post-injury was seen as advantageous to the state in reducing the long-term reliance on the state for disability or unemployment grants. Various parties within society were additionally seen to have a role to play in accommodating the injured patient and assisting them back into productivity.

*"The state benefits because, again depends on what the state capacity is for providing disability support. If one reduces disability and improves outcome by doing a better job in the acute care scenario, as well as with rehab services, you reduce the overall burden on the state"* [O]

*"For example if they have traumatic brain injury and we have a patient who develops seizures or this patient lose some physical capabilities if they follow, if I give the assistance for a longer period I think that the chance of having better results in terms of rehab is bigger. So the consequence of this is that the probability of return to work is bigger if they have a closer medical assistance for a longer period."* [J]

*"if you have the long-term outcomes with you, you will be in a much better position to know—encourage people to follow road safety rules, not to drink and drive, not to do this, not to do that. So, in order to make any guidelines, you need to have long-term outcomes [. . .] all these long-term follow-ups have benefits-one is for the policymakers, one is for the patient. [. . .] Then you can tell where you're falling short, what do you need to improve and how much is the input required by each one of the players in the game- you know what I mean? Stakeholders. One is the system, the society, the police officers, doctors, nurses, and so on, and so forth."* [N]

Governments, state officials, insurance and legislative bodies were deemed to have a responsibility in both ensuring and providing the means for follow-up, described as 'state buy-in', whereby the state should contribute to the investments and resources required, including: equipment, transport services and adequate post-graduate training programmes.

**Science.**    Follow-up was also considered advantageous for science, where local data contributed to the international evidence base on TBI in low-resource settings.

*"We can recognise the factors in long-term follow-up that can be minimised. The treatment protocols, like if you are giving one treatment [. . .] we follow the patient for long-term so we can do research and learn many things, like treatment protocols, like what treatment is effective, which treatment is ineffective."* [E]

*"most of times we need long-term follow up for mainly to try out researches and the follow up of this patient in outpatient system [. . .] [there] is a benefit to our society because the production or scientific production is a way that we have to go through in order to improve our universities to improve our medical team. So having, having this follow up we can make science in a better way in this helps us to, to improve."* [J]

*"It's good for research because we should have data of our patients. We should save everything about them to see what we can add more in our practice or we should to improve in our practice"* [M]

A few regarded research as a key driver in conducting follow-up, with trials, cohort studies and registries all facilitating the collection of reliable data, to improve local practice.

## Lost to follow-up: Who misses out and why?

Theme four considers the many reasons a patient may become lost to follow-up and describes: the absent patient; the recovered patient; and the triaged patient.

**The absent patient.**    Poverty was regarded as the most substantial barrier to follow-up, with many patients described as 'financially exhausted', their socioeconomic status largely governing their post-injury care. With the centralisation of many specialist services and lack of suitable or affordable transport options, geographical barriers prevented many from accessing

outpatient services. Disability and assistance needs exacerbated perceived barriers. Participants felt that communication often broke down after discharge to the community, where patients were unable to maintain contact with their clinician, or new care providers lacked appropriate channels of communication with the neurotrauma hospital. Therefore, follow-up was seen by a few clinicians as a privilege, reflecting a lack of equal access to all that may benefit.

*"the centre where I am working is a tertiary centre. And it's in the capital city. So those patients who are outside, out of the [city], so there are for them sometimes it will be difficult for them to come. [. . .] Because it will be very far. It takes one day to come by bus [. . .] that will be challenging. Sometimes the patient do not come to follow up because of the distance"* [B]

*"In our country it is very difficult to do the following process in person face to face because due to the organisation of the system, many patients will never return after surgery or they return only in a short period of time like one or three months after surgery but then they are allocated by the insurance systems in very different places. So, most of these places never or rarely didn't belong to the same centres that are doing the acute care so they belong to some other insurance areas and the communication is totally lost."* [I]

*"Sometimes also you know like patients who have- patients have sometimes psychological problems regarding their disabilities. So they are frustrated when they come to the hospital. I have seen some sort of that patient. So they don't come. They prefer to stay at home. Some people also go to private clinics so they have missed follow ups at outpatient clinic of [the] university hospital."* [G]

Participants also thought patients and families were not always aware of TBI sequalae and the value of follow-up to recovery. It was also these more subtle sequalae that some participants felt may be the reason a patient did not attend for follow-up. These sequelae were often thought to be psychological, behavioural or cognitive in nature, and were perceived to manifest in a patient forgetting appointments, refusing medication, or lack of willingness to cooperate with further follow-up, or a general lack of concern for their own recovery.

**The recovered patient.** Participants felt patients and families may consider recovery to be complete, or complete enough, to not require further follow-up services. Some suggested that patients would be in contact if there were genuine need. Therefore, a lapse in contact was often interpreted as the patient reaching a satisfactory level of recovery.

*"some of them do not come after one month if they feel that they are doing okay, they don't come is a like a problem of culture but sometimes they don't come and they come with complications that is a bad thing for the kind of patient we do deal with like in Africa it's a problem of culture"* [K]

*"But then [the patient] starts feeling that he is well, he is performing daily activities, he's not able to concentrate on minute activities or major functional activities. And his family members are also, "okay, okay that's good, he is taking himself and he is going and". They're not concerned about the behaviour changes or the complexity of activity which he is unable to perform after head injury or so. So once that patient starts feeling that he's better and the relatives of the patient start feeling he is better, their patient is better, okay. But then they feel there is no need to follow up."* [P]

However, concerns were raised by a few participants that patients and their families may be more focused on physical recovery and less on the more subtle sequelae such as psychological or cognitive changes that would equally affect their daily lives.

**The triaged patient.**   The dominant view of clinicians was that multiple instances of long-term follow-up was both needed and advantageous, yet contrasting views suggested follow-up may not be for everyone, and was dependant on the accessibility, capacities, and objectives of the care provider, and the wishes of the patient.

*"I guess that a longer follow up may be, may be good in some cases but not in all cases and again I would say that it is the new era I don't think that a formal follow up will be, could be like I guess that a longer follow up can be replaced with connecting with through the Internet or through the phone call. So I think that only in a few cases would qualify for a longer follow up"* [D]

*"Also to specify the period of follow up for each patient, so that I can't exhaust myself in someone, someone who was having like mild non-surgical, small extradural haematoma, he shouldn't come to follow up after six months. It won't benefit anything, okay. But someone who is brain contusion and has neurological deficits, he should come to assess if he can continue on rehabilitation, if he's improving or not. Because sometimes it takes about six months until he moves his limb. Okay? So we should weigh those who can come for follow over long time, and those who can come for follow up over short time."* [G]

*"Because we cannot treat everybody optimally. We don't have the facilities. [. . .] So for trauma, trauma follow up is prognostication. I can look after this guy, this guy there's no point. This one there's a point in doing that kind of stuff."* [F]

Severe cases or patients with continuing complications were prioritised, those with no further problems were less important for follow-up.

*"So, we routinely follow up severe TBI children for a period of about four to five years. So that's run by our surgical department. But we connect with developmental services, rehab services, speech therapy, occupational therapy, physiotherapy. [long-term follow-up is for] the severe cases and the ones with moderate TBI who flag up problems. So, it all depends really on what the assessment in the early phase is."* [O]

## Processes and procedures: What do we do, or what might we do?

The final theme considers the current methods of in-person follow-up, technology-based follow-up, and the perceived barriers and additional considerations for follow-up as a whole.

**In-person assessment.**   Participants emphasised the value of in-person assessments which facilitated physical assessment and were thus considered more reliable than remote methods. However, in the context of overburdened services, assessments were often non-specific and focused on managing any further physical manifestations of injury. Due to time pressures few opportunities existed for more sensitive outcome assessment beyond a binary 'fit and well, or not'.

*"we bring them in. [. . .] we always physically see them. We tend not to rely on remote measures in part because it's a lot less predictable whether- if they've got capacity for kind of internet capacity for instance is going to be much reduced in our communities. [. . .]in those exceptional cases we'll try to make use of kind of remote assessment or link them in with a*

*local hospital and then communicate with that hospital. But for the vast majority of our patients we try to get them physically in, because that's the most reliable way of doing so. [. . .] I think it's hard to give you one answer across all the tools. I think that it depends on the quality of initial assessment that was done. So, the primary thing that one's losing with online tools is your ability to physical examine a patient." [O]*

*"I think it is better to see the patient like that we can make a neurological exam correctly, but in the case that we don't have choice yes it is better to use telephone to know if, to ask some questions for the patient and to get feedback about that you know if they're going well or not. Yeah but it is not sufficient but if we don't have choice yes it is better to telephone them" [M]*

*"We see the general condition of a patent, any neurological deficiency the patient is having, okay. And we are unable to perform the full battery of tests okay which will be required to know every lobe signs, okay. So, the amount of time which one has to give on a neurosurgical patient at least on the first follow up is decreased." [P]*

*"In the assessment we do, I say to you that the time is short for each patient. Once the patient comes, I look at the card of the patient. I see the initial diagnosis, what happened to him inside our department. And the last follow up. Then I ask the patient is there any new event or problem? If he says no, it's okay, no problem. I look at the medications and I see if- I check for the medication he will go on. And I check for the medications that he should stop. That's for a regular patient. If the patient has a problem when I ask him, I examine the patient, it's just a crude examination. It's not a full neurological examination. Because there is no time." [G]*

Where used, the Glasgow Outcome Scale, its Extended version, and the Modified Rankin Scale were the most common outcome measures completed during in-person assessment, providing clinical teams with a reliable record of patient outcome. Other outcomes used by some included the Karnofsky scale, Lagos Brain Disability Examination Scale, Mayo-Portland Inventory, and 36-Item Short Form Survey.

**Technology-based assessment.**   Despite the endorsement of in-person assessments, almost all participants felt incorporating technologies would prove useful in follow-up and many had begun to use remote technology to meet patient needs during the COVID-19 pandemic. Technology to enhance follow-up was also part of a wider desire for improved infrastructure including electronic patient records and appointment management systems.

SMS, telephone, and videoconferencing were positively appraised. However, some technologies were favoured over others, suggesting a hierarchy existed. Telephone and instant messaging were particularly favoured due to their accessibility, ability to replicate in-person assessment and suitability for those with limited education or literacy. Advantages to patients included increasing access to follow-up and reducing unnecessary travel. Burden on outpatient facilities was also thought to be reduced by dealing with concerns swiftly by remote means.

*"Just get their number, get their mobile phone number. The telephoning density mobile system especially is so remarkably good, even where you think people are so down-and-out, and they have it, mobile phones." [R]*

*"Yeah, one is obviously telephone and if the telephone can be added with videos, like a patient can show you different parts of their bodies or specific examination and show you whether they are able to move their limbs, these kind of things. Or if they have wound, they take a picture of it and take videos and send to you, that you can analyse it, these kind of things help." [Q]*

*"Most of the time when we cannot observe a seizure we ask our patient to send us a video when the patient seizures. So many patients, many attendants do send us a video of their patients when they experiencing the seizure. So in that case we have a very good idea then what's going on. This is something that I think that the use of WhatsApp is very good in that case."* [D]

*"So, some of our patients will be travelling for four to six hours just to come to a clinic appointment with me that's over in 15, 20 minutes, and they get back on their transport to go back. Horribly, horribly inefficient way of running an economy. So, for a relatively modest investment in online support locally, whether for each individual family, or some kind of hub, massive savings and massive reduction in inconvenience to patients."* [O]

Additional benefits included opportunities for more frequent points of patient contact, in addition to providing infrastructure for connected inter-hospital systems and providing formal outcome data that could be used in research to improve practice.

**Barriers and additional considerations for follow-up.** Technology was seen by some as primarily for research and not every day clinical practice. Additional costs for personnel, infrastructure, or equipment to deliver such technology was hard to justify as was any intervention that may increase the pressure on an overburdened health service. Patient executive functioning, language and literacy were also considered barriers to effective assessment using remote means and may require reliance on proxies such as family members.

*"We should utilise any technologies which will enable us to have periodical or continuous assessments without lead[ing] to a heavy toll on healthcare workers."* [H]

*"I will suggest to try to have a lot of neurosurgeons like that. Maybe some of us could do like, I don't know like a boot camp in the, in the rural town to go to try to see the patients that can just walk, to come to see this boot camp in the area. That's maybe we should have agreed marking about our patients in the region. Like that we can try to make an organisation that yes you know this month you will go in this area to try to follow up of the TBI patients that we take care something like that"* [M]

*"Cost is most like, it should be done on that government sector level. There should be more hospitals, more neurosurgical training centres, and more funds allocated to research [. . .] That should be going into each and every hospital even by the government [. . .] there should be some sort of transportation for these patients. So if transportation is available, I think it will be much easier to follow those patients"* [E]

Lastly, several secondary considerations were identified as relevant to improved long-term follow-up in LMICs. These included: improved long-term neurotrauma services, more neurosurgical training programmes, peripheral or satellite neurotrauma sites, increased personnel, improved transport infrastructure, and increased funding for local research.

## Discussion

This study is the first to examine neurotrauma physicians' perspectives of the contextual challenges of long-term follow-up following TBI in low-resource settings. Our findings reflect how the inequalities in healthcare affect the provision of long-term follow-up. While neurotrauma care is centralised to densely populated cities [53], much of the population reside in rural areas where specialist neurotrauma care is not available. Corresponding with the literature, patient characteristics including geography (or rurality), poverty, low levels of education and literacy

were described as causative factors in losses to follow-up, which further marginalises or exacerbates social inequality to TBI patients in these settings [54].

Where patients can reach the clinic, in-person assessment emerged as often crude and physically oriented. A 'survival-first' model of care was perceived to exist, with a tension between clinical time, obvious physical needs, and more subtle sequelae that required more sensitive and complex intervention. Rehabilitative services were also seen to be in very limited supply. Where available, they were often separated from the neurotrauma care provider and hosted within the private sector. This is in stark contrast to the UK service commissioning guidelines that calls for integrated, long-term rehabilitation across all phases of care, co-ordinating across geographical regions where complex needs cannot be met by local services [55]. Additional insights may be garnered from further research examining the working relationships and pathways between the acute neurosurgical and longer-term rehabilitative services in LMIC settings.

Our findings suggest the reality of long-term follow-up in LMICs is often limited to six or 12 months post-injury which contrasts to definitions of long-term follow-up in HIC-originating literature [56–58] which define long-term as five years and beyond. However, due to the limited resources follow-up beyond one year may simply be unfeasible for those in LMICs. Similarly, those in HICs may be limited where insurance schemes do not routinely cover longer period of follow-up. This may account for follow-up beyond 12 months being perceived as associated with research rather than everyday practice. The importance of research to health is undisputed; by increasing research capacity, the health outcomes of disadvantaged patients may also be improved [59]. However, the benefits of regular, long-term follow-up has been challenged in a recent paper from India that suggests that, in developing countries lacking rehabilitative facilities, patient outcome at discharge in severe cases of head injury may be representative of their final outcome overall [60]. In contrast, another study between India and the United States examined ICU guideline adherence and inpatient mortality in severe TBI, and demonstrated that, although long-term outcomes generally improved upon discharge, patients discharged with favourable outcomes often still deteriorated in the community [61].

Our findings also indicate the potential for remote follow-up technologies to improve follow-up services and reduce loss to follow-up. Small's [62] definition is useful here "Lost to follow-up means more than loss—it means failure to find". Technology-assisted follow-up brings with it several opportunities to track and locate patients post-discharge. However, several factors must be considered including universal access and cost, technology, and networking availability [63], user education and literacy [63, 64], and cultural values. Technology-based solutions must exist to reduce rather than exacerbate social health inequalities [54], in order to bring specialists to patients in the remotest corners of low-resource settings.

Our findings mirror those of others regarding the challenges of using remote follow-up technologies [53], these include concerns about the availability of technology, initial investments and maintenance costs, network coverage, and additional personnel required to facilitate these services. Technologies should not further encumber overburdened clinical teams. In low-resource settings these are crucial considerations to successful implementation.

Although a one-solution-fits-all approach may not be possible, our findings suggest technologies should be 'common-denominator' such as telephone and SMS. Audio-visual technologies (e.g., telephone, videoconferencing) may enable more sensitive follow-up in a manner closer to in-person visits, although use of such technology warrants further investigation.

Amongst the challenges of long-term outcome data collection following hospital discharge, obstacles still persist in adequate incidence data capture in low-resource settings, with the global TBI incidence expected to be far greater than current estimates [9, 65, 66]. Although

beyond the scope of our inquiry, future research may consider the causes of, and solutions for improving, TBI incidence reporting so that the full magnitude of TBI is understood.

Both direct and indirect costs arise from acute care and rehabilitation, adding to the narrative that finance remains a stalwart barrier to equitable surgical care [67]. Social and economic consequences of TBI are often long-lasting [68]. Neurosurgeons felt patients made cost-based decisions on what post-hospital services they chose to access [69, 70]. However, this assumption warrants further investigation from the patient perspective.

Our findings may also help us to understand follow-up practices in HICs, and what may be compatible solutions in these settings. Although vast differences exist between these settings with respect to infrastructure, personnel and resources [71], similarities exist with a lack of suitable and consistent injury surveillance and reporting systems [21]. Technology-based outcome collection may then facilitate improved surveillance globally, filling voids in epidemiological data and building bridges between patient and physician following discharge.

## Study recommendations

From the findings of this study, we offer several practice and future research recommendations.

Participants perceived that long-term follow-up in low-resource settings is in the interests of many, from patient to physician, to state and science; harmonious with the WHO's promotion of surveillance and outcome data collection for the reduction of the global burden of TBI. Despite these benefits, numerous challenges exist to access, deliver, and retain follow-up services in the long-term, and the importance of follow-up was perceived by physicians as not always recognised by patients and families. We recommend that patients and their families are educated about the specific value of follow-up for the individual's recovery and re-integration into the wider community. For long-term follow-up to be useful, it must be linked predominantly with the capacity to offer further therapies and interventions for the improved outcomes of patients and must not merely be an avenue for data collection for epidemiological or surveillance research. Further studies are warranted to evaluate the contextual challenges of follow-up from the patient's perspective.

This study showed that whilst physicians were mostly aware of the multifaceted benefits of follow-up, the constraints of an over-burdened healthcare system mean that comprehensive follow-up post-discharge may be simply not possible. It is therefore recommended that technologies are introduced with the specific aim of reducing the clinical burden on the physician, whilst also increasing the reliability and consistency of follow-up data in LMICs. Future research is required that provides a detailed evaluation of alternative technologies and the impact these have on the clinical care team, including the views of allied health professionals and the wider neurorehabilitation workforce.

In the context of LMICs, where TBI exacerbates high levels of poverty both for patients and health services, technology may offer an alternative to in-patient assessment both geographically and financially beyond reach of many. Therefore, when creating and recommending technologies, a careful economic evaluation from the position of patients and health services is recommended, in addition to being cognisant of the level of literacy and capabilities of the end-user.

## Study limitations

This study did not reach its target sample size and is therefore limited in its ability to reflect the heterogeneity within LMICs. However, determining qualitative sample size *a priori* is problematic and, as previously discussed, data saturation is contentious in reflexive thematic

analysis [37]. As such, generalisability of findings is not typically sought nor is regarded to be construed from the study findings [72]. In addition, these findings were limited to the views of neurosurgeons, examining the experiences of those working in rehabilitation services may lead to a more comprehensive understanding of long-term follow-up in LMICs. Finally, findings are influenced by the lens of the lead authors and further studies led by those within LMICs are recommended. However, despite these limitations this study is the among the first to portray a complex issue of importance to the global neurosurgical community that will resonate with others.

## Conclusion

The collection of long-term outcome data plays an imperative role in reducing the global burden of neurotrauma. Therefore, this was an exploratory study that examined the contextual challenges associated with long-term follow-up in LMICs. Whilst physicians were mostly aware of the multifaceted benefits of follow-up, comprehensive follow-up within the constraints of an over-burdened healthcare system may simply not be possible. Where technology can contribute to improved neurotrauma surveillance and remote assessment, offering an alternative to in-patient assessment geographically and financially out of reach for many, these must be implemented in a manner that improves patient outcomes, reduces clinical burden on physicians, and does not surpass the comprehension, capabilities, or financial means of the end user. Future research is recommended on patient and family perspectives, the impact on clinical care teams, and the full economic implications of new technologies for follow-up.

## Supporting information

**S1 File.**
(DOCX)

**S1 Checklist.**
(DOCX)

## Acknowledgments

Abenezer Tirsit Aklilu, Amos O. Adeleye, Bhagavatula Indira Devi, Anthony Figaji, Ankur Bajaj, Muhammad Tariq, Pritam Gurung, Tsegazeab Lake, Andres Mariano Rubiano, Ehanga Idi Marcel, Claire Karekezi, Nourou Dine Adeniran Bankole, Liew Boon Seng, Olufemi Emmanuel Idowu, Gustavo Sousa Noleto, Noor-ul-Huda Maŕia, Mohammad A Azab.

## Author Contributions

**Conceptualization:** Brandon G. Smith, Charlotte J. Whiffin, Peter J. Hutchinson, Angelos G. Kolias.

**Data curation:** Brandon G. Smith.

**Investigation:** Brandon G. Smith, Charlotte J. Whiffin.

**Methodology:** Brandon G. Smith, Charlotte J. Whiffin, Tom Bashford.

**Project administration:** Brandon G. Smith.

**Software:** Brandon G. Smith.

**Supervision:** Charlotte J. Whiffin, Peter J. Hutchinson, Angelos G. Kolias.

**Writing – original draft:** Brandon G. Smith.

**Writing – review & editing:** Brandon G. Smith, Charlotte J. Whiffin, Ignatius N. Esene, Claire Karekezi, Tom Bashford, Muhammad Mukhtar Khan, Davi J. Fontoura Solla, Bhagavatula Indira Devi, Wellingson S. Paiva, Franco Servadei, Peter J. Hutchinson, Angelos G. Kolias, Anthony Figaji, Andres M. Rubiano.

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
