## [Decision Letter · Decision Letter 0]

12 May 2022

PONE-D-22-03840Neurotrauma clinicians’ perspectives on the contextual challenges associated with traumatic brain injury follow up in low-income and middle-income countries: a reflexive thematic analysisPLOS ONE

Dear Dr. Smith,

Thank you for submitting your manuscript to PLOS ONE. After careful consideration, we feel that it has merit but does not fully meet PLOS ONE’s publication criteria as it currently stands. Therefore, we invite you to submit a revised version of the manuscript that addresses the points raised during the review process.

We look forward to receiving your revised manuscript.

Kind regards,

Gregory W.J. Hawryluk, MD, PhD, FRCSC

Academic Editor

PLOS ONE

Journal Requirements:

2. In your Methods section, with reference to the use of Twitter and WhatsApp, please include additional information about your dataset and ensure that you have included a statement specifying whether the collection and analysis method complied with the terms and conditions for the source of the data

3. Please include a complete copy of PLOS’ questionnaire on inclusivity in global research in your revised manuscript. Our policy for research in this area aims to improve transparency in the reporting of research performed outside of researchers’ own country or community. The policy applies to researchers who have travelled to a different country to conduct research, research with Indigenous populations or their lands, and research on cultural artefacts. The questionnaire can also be requested at the journal’s discretion for any other submissions, even if these conditions are not met.  Please find more information on the policy and a link to download a blank copy of the questionnaire here: https://journals.plos.org/plosone/s/best-practices-in-research-reporting. Please upload a completed version of your questionnaire as Supporting Information when you resubmit your manuscript

"AGK and PJAH are supported by the National Institute for Health Research (NIHR) Cambridge Biomedical Research Centre, Royal College of Surgeons of England, and the NIHR Global Health Research Group on Neurotrauma. PJH is also supported by an NIHR Research Professorship. The NIHR Global Health Research Group on Neurotrauma was commissioned by the UK NIHR using Official Development Assistance funding (project no. 16/137/105). INE, CK, MMK, DJFS and AGK are members of the Young Neurosurgeons Committee of the World Federation of Neurosurgical Societies. The committee is supporting this project."

We note that you received funding from a commercial source: National Institute for Health Research (NIHR) Cambridge Biomedical Research Centre

Please include your amended Competing Interests Statement within your cover letter. We will change the online submission form on your behalf

Additional Editor Comments (if provided):

My apologies again for the delays inherent to the review process. The three reviewers were in agreement that your manuscript shows promise for publication. I would ask that you please carefully onsider their requests for minor revision and provide point-by-point responses to their critiques. We look forward to reviewing a revision of your work.

Thank you!

Reviewers' comments:

Reviewer's Responses to Questions

**Comments to the Author**

1. Is the manuscript technically sound, and do the data support the conclusions?

Reviewer #1: Yes

Reviewer #2: Yes

Reviewer #3: Yes

2. Has the statistical analysis been performed appropriately and rigorously? 

Reviewer #1: N/A

Reviewer #2: N/A

Reviewer #3: N/A

3. Have the authors made all data underlying the findings in their manuscript fully available?

Reviewer #1: Yes

Reviewer #2: Yes

Reviewer #3: Yes

4. Is the manuscript presented in an intelligible fashion and written in standard English?

Reviewer #1: Yes

Reviewer #2: Yes

Reviewer #3: No

5. Review Comments to the Author

Reviewer #1: Qualitative research is not commonly undertaken in the field of neurosurgery and yet an increasing number of papers have been published over the last several years which use this labor-intensive methodology to advance our understanding key issues in a field from a person-centered approach. This paper applies such methodology to developing an understanding of long-term follow-up of traumatic brain injury patients in low resource settings. After interviewing 18 neurosurgeons from 13 different countries the authors identified 5 main themes and discuss their findings in the context of managing these patients and providing future direction for both research and care.

Overall, this is a well written paper that is grounded in solid methodology. Its findings are interesting and relevant to the emerging conversation in global neurosurgery which is struggling to define the scope and depth of challenges in neurosurgical care in LMICs due largely to the limitations of collecting data in the traditional sense. The methodology section is somewhat lengthy as the authors spend a great number of words explaining many details to presumably enlighten a readership that is naïve to such approaches. They might do well to shorten the “Data Analysis” and “Rigour” sections and let those readers who take an interest in “Reflexive Thematic Analysis”, for example”, to pursue it elsewhere. Beyond that I think their paper needs little modification. And, for what it is worth, I believe that the authors conclusion that further studies of this topic from the patients’ perspective are needed cannot be overstated enough; it is in these data that we be needed to truly advance the treatment of neurotrauma patients in LMICs.

Reviewer #2: Congratulations for the study and manuscript, all PLOS ONE guidelines have been met.

My only recommendation is about exclusion criteria, those were rightly opposite to the inclusion criteria, thus, they are unnecessary.

Reviewer #3: This is an interesting point addressed by the authors due to the lack of discussion regarding the challenges faced by LMIC researchers, when it comes to data collection and follow up of TBI patients. The authors performed a qualitative analysis of the perceptions of researchers from LMIC and found 5 themes. However, there are some aspects that need to be better clarified in the methods section (particularly regarding the excess of explanation and lack of how each phase were done). Moreover, as this is a very “dense” paper, the writing must be improved for clarity and conciseness (i.e, in lines 99 and 120, it is missing a comma; there are several wordy sentences like “the aim of inquiry “ , “through interpretation of”, “additionally at this time”, etc; there are lots of passive voice along the text – lines 111, 112, 119, 157, 189 etc).

Find bellow specific comments:

Abstract

In the conclusion: change “results in improved” to “improves”

Conclusion is confused (rewrite sentence for clarity)

Introduction

First two sentences: be more specific, showing quantitative data.

Consider rewrite this sentence for clarity: “Accurate data also facilitates the evaluation of systems and services, enabling: identification targets of wider systems improvement in injury prevention,(15) assessment of the efficacy of management decisions and patient treatment,(10,14) facilitation of quality improvement initiatives and clinical trials, and lastly, the establishment and continuation of registries; of which themselves facilitate the development of care pathways, health policy, and injury prevention strategies” - (maybe, taking of “…of which themselves… strategies”and/or splitting in two sentences)

Consider taking off “that does exist” of the sentence in lines 81 and 82

Methods

In line 111 and 113: change “may” to “might”

Typo in line 115“renumerate”, and missing comma after “participants”:

In lines 148-153: Hard to read. Consider splitting the sentence…

In line 174: please specify the demographic data collected

In lines 181-184: Hard to read sentence

Was the semi- structured interview guide pilot tested?

I think it’s better to put the initials of the authors along with the text in the following format: “B.G.S.” instead of “BGS”

Sometimes along with the methods, the authors explain and discuss about the methodology. I suggest that the authors stick to describing the methods and not justifying it. The authors published the protocol (BMJ, 2021), therefore, you might use it to save words.

It was hard to understand what the authors wanted to say with this sentence: “Through the use of NVivo and reflexive journal writing, a clear audit trail was created from data to findings, which also adds to the dependability and confirmability of the study”. It would be interesting if the authors could explain how they used the NVivo and the reflexive journal writing to guarantee rigour.

Moreover, consider providing the codebook and a sample analysis of interview transcripts as supplemental material.

Results

In line 239, take the word “Unfortunatelly” off.

Please specify all the reasons for participants withdrawing from the study (and quantify objectively)

In table 2: Years of experience – 1 to 5 or 2 to 5? I thought one year of experience was an exclusion criteria.

It is interesting to be persuasive when writing the paper, however, I think it is too much. Consider taking off some words such as “important” or “illuminate”.

The themes are well presented and highlight several issues related to the low-resource environment challenges.

Discussion is well written

Conclusion

Focus on the leading suggestions derived from the study.

It is the same as the abstract. Consider being more concise in the abstract.

Finally, I’d like to congratulate the authors for their hard work and shed light on this issue.

6. PLOS authors have the option to publish the peer review history of their article (what does this mean?). If published, this will include your full peer review and any attached files.

Reviewer #1: No

Reviewer #2: **Yes: **Sérgio Brasil

Reviewer #3: **Yes: **ROBSON LUIS OLIVEIRA DE AMORIM

---

## [Author Response · Author response to Decision Letter 0]

1 Jul 2022

Journal / Editor 1

Thank you for your comment - we have since amended all Tables to bring them in line with PLOS One Table formatting guidelines

Location: Tables 1 - 3

In your Methods section, with reference to the use of Twitter and WhatsApp, please include additional information about your dataset and ensure that you have included a statement specifying whether the collection and analysis method complied with the terms and conditions for the source of the data

Thank you - we have added further detail regarding how potential participants expressing interest via social media platforms were converted to e-mail for further discussion. A terms and conditions compliance statement has also been added.

Location: Pg. 6

Please include a complete copy of PLOS’ questionnaire on inclusivity in global research in your revised manuscript. Our policy for research in this area aims to improve transparency in the reporting of research performed outside of researchers’ own country or community. The policy applies to researchers who have travelled to a different country to conduct research, research with Indigenous populations or their lands, and research on cultural artefacts. The questionnaire can also be requested at the journal’s discretion for any other submissions, even if these conditions are not met. Please find more information on the policy and a link to download a blank copy of the questionnaire here: https://journals.plos.org/plosone/s/best-practices-in-research-reporting. Please upload a completed version of your questionnaire as Supporting Information when you resubmit your manuscript

Thank you for this suggestion. We have completed the Inclusivity in global research questionnaire as suggested for inclusion as supplementary material, appending a reference of this to our Methods section.

Location: Supplementary file S1, Pg. 14

We note that you received funding from a commercial source: National Institute for Health Research (NIHR) Cambridge Biomedical Research Centre

Thank you for your comment, however, the National Institute for Health Research (NIHR) Cambridge Biomedical Research Centre is funded by the U.K. Government, and is not a commercial source.

Full funding details of the NIHR Cambridge Biomedical Research Centre can be found at https://fundingawards.nihr.ac.uk/award/IS-BRC-1215-20014.

Please review your reference list to ensure that it is complete and correct. If you have cited papers that have been retracted, please include the rationale for doing so in the manuscript text or remove these references and replace them with relevant current references. Any changes to the reference list should be mentioned in the rebuttal letter that accompanies your revised manuscript. If you need to cite a retracted article, indicate the article’s retracted status in the References list and also include a citation and full reference for the retraction notice.

Thank you for your suggestion - we have ensured the reference list is complete and correct.

I can confirm that we are not aware of any retracted papers within our references.

Qualitative research is not commonly undertaken in the field of neurosurgery and yet an increasing number of papers have been published over the last several years which use this labor-intensive methodology to advance our understanding key issues in a field from a person-centered approach. This paper applies such methodology to developing an understanding of long-term follow-up of traumatic brain injury patients in low resource settings. After interviewing 18 neurosurgeons from 13 different countries the authors identified 5 main themes and discuss their findings in the context of managing these patients and providing future direction for both research and care.

Overall, this is a well written paper that is grounded in solid methodology. Its findings are interesting and relevant to the emerging conversation in global neurosurgery which is struggling to define the scope and depth of challenges in neurosurgical care in LMICs due largely to the limitations of collecting data in the traditional sense. 

Thank you - we are grateful for your positive comments.

The methodology section is somewhat lengthy as the authors spend a great number of words explaining many details to presumably enlighten a readership that is naïve to such approaches. They might do well to shorten the “Data Analysis” and “Rigour” sections and let those readers who take an interest in “Reflexive Thematic Analysis”, for example”, to pursue it elsewhere 

Thank you for your comments. We have amended our Data Analysis & Rigour sections, and we now believe it provides a more succinct account of our methods.

Location: Pgs. 9 - 14

Congratulations for the study and manuscript, all PLOS ONE guidelines have been met.

My only recommendation is about exclusion criteria, those were rightly opposite to the inclusion criteria, thus, they are unnecessary.

Thank you - we are grateful for your positive comments. We agree that exclusion criteria in this instance Is superfluous and have since removed it. We wish to note that a limitation of Microsoft Word does not allow this removal to appear as a tracked change; we trust this reference appropriately acknowledges this modification. 

Location: Table 1 (Pg. 6)

This is an interesting point addressed by the authors due to the lack of discussion regarding the challenges faced by LMIC researchers, when it comes to data collection and follow up of TBI patients. The authors performed a qualitative analysis of the perceptions of researchers from LMIC and found 5 themes.

Thank you for your comment.

However, there are some aspects that need to be better clarified in the methods section (particularly regarding the excess of explanation and lack of how each phase were done).

Thank you for your comment - in line with comment 7, we believe that this revised manuscript section offers a clarified and succinct description of our methods.

Location: Pgs. 9 - 14

Moreover, as this is a very “dense” paper, the writing must be improved for clarity and conciseness (i.e, in lines 99 and 120, it is missing a comma; there are several wordy sentences like “the aim of inquiry “ , “through interpretation of”, “additionally at this time”, etc; there are lots of passive voice along the text – lines 111, 112, 119, 157, 189 etc).

Thank you for your comments - we have made additional modifications to the style and phrasing to improve the readability throughout the manuscript, and in line with previous comments, believe that this revised manuscript is now more succinct and clearer for the reader.

Location: Pgs. 9 - 14

Abstract In the conclusion: change “results in improved” to “improves”

Conclusion is confused (rewrite sentence for clarity)

Thank you for your comment - we hope to have addressed this issue of clarity within our abstract and conclusion.

Location: Pgs. 2, 35

Introduction

First two sentences: be more specific, showing quantitative data. 

Thank you for your suggestion - we have now added figures from Dewan et al. pertaining to global estimates of TBI incidence. We hope this serves as a useful reference point that details the scale of this global health concern, with LMICs disproportionately affected.

Location: Pg. 3

Consider rewrite this sentence for clarity: “Accurate data also facilitates the evaluation of systems and services, enabling: identification targets of wider systems improvement in injury prevention,(15) assessment of the efficacy of management decisions and patient treatment,(10,14) facilitation of quality improvement initiatives and clinical trials, and lastly, the establishment and continuation of registries; of which themselves facilitate the development of care pathways, health policy, and injury prevention strategies” - (maybe, taking of “…of which themselves… strategies”and/or splitting in two sentences) 

Thank you for this suggestion - we agree that splitting the sentence in two adds clarity and have implemented this

Location: Pg. 3

Consider taking off “that does exist” of the sentence in lines 81 and 82 

Thank you for your comment - we have removed this phrase to improve clarity.

Location: Pg. 3

Methods. In line 111 and 113: change “may” to “might”

Typo in line 115“renumerate”, and missing comma after “participants”:

Thank you for these comments - we have modified our ethics paragraph to reflect these changes.

Location: Pg. 5

In lines 148-153: Hard to read. Consider splitting the sentence…

Thank you for this suggestion - we have divided the structure of this paragraph into smaller, more comprehensible sentences.

Location: Pg. 7

In line 174: please specify the demographic data collected 

Thank you for your comment - we have specified the data collected.

Location: Pg. 9

In lines 181-184: Hard to read sentence 

Thank you for this suggestion - we have modified the sentence structure to enhance clarity.

Location: Pg. 10

Was the semi- structured interview guide pilot tested? 

Thank you for your comment. Though no separate pilot study was conducted, after three interviews, the interview guide was reviewed with co-authors and critically discussed with respect to its administration, content, and researcher interview style. No revisions were made to the protocol or guide at this stage, and feedback relating to interview style was incorporated into further interviews. We have added this additional information to our methods.

Location: Pg. 9

I think it’s better to put the initials of the authors along with the text in the following format: “B.G.S.” instead of “BGS” Thank you - we have corrected the format of all initials. Pgs. 9, 10, 12, 13

Sometimes along with the methods, the authors explain and discuss about the methodology. I suggest that the authors stick to describing the methods and not justifying it. The authors published the protocol (BMJ, 2021), therefore, you might use it to save words. 

Thank you for your comment - in line with other suggestions, we have taken this opportunity to modify our methods section, and now believe this provides a more succinct account of the works undertaken.

Location: Pgs. 9 - 13

It was hard to understand what the authors wanted to say with this sentence: “Through the use of NVivo and reflexive journal writing, a clear audit trail was created from data to findings, which also adds to the dependability and confirmability of the study”. It would be interesting if the authors could explain how they used the NVivo and the reflexive journal writing to guarantee rigour. 

Thank you for your comment - we have provided further clarity regarding our use of NVivo software in the creation of an audit trail.

Location: Pgs. 13, 14

Moreover, consider providing the codebook and a sample analysis of interview transcripts as supplemental material. 

Thank you for this suggestion, which we have discussed at great length. Reflexive thematic analysis is not a ‘codebook’ approach in the same way as something like framework analysis (see Braun, V. and Clarke, V., 2021. Can I use TA? Should I use TA? Should I not use TA? Comparing reflexive thematic analysis and other pattern‐based qualitative analytic approaches. Counselling and Psychotherapy Research, 21(1), pp.37-47). 

Therefore, the analysis and process of moving from codes to themes is not formulaic and does not rely on a fixed (often a priori) code book that other techniques may employ. In reflexive thematic analysis there is no direct route from codes to the interpretation that can be laid bare through the inclusion of a codebook or sample analysis and as such would have little value to the reader. We acknowledge that the COREQ checklist suggest including this; however, these recommended items are being questioned in recent literature (see Buus, N. and Perron, A., 2020. The quality of quality criteria: Replicating the development of the Consolidated Criteria for Reporting Qualitative Research (COREQ). International Journal of Nursing Studies, 102, p.103452). 

Therefore, we have not provided a codebook or sample analysis in this manuscript and hope our reasons for this are acceptable, however we have added further information pertaining to coding and theme development in our methods. 

Location: Pg. 10

Results. In line 239, take the word “Unfortunatelly” off.

Thank you for this suggestion - this has been removed.

Location: Pg. 14

Please specify all the reasons for participants withdrawing from the study (and quantify objectively)

Thank you for your comment - to clarify, of the 55 individuals expressing interest 19 participants were consented, however one participant withdrew (by default) as contact was lost prior to conducting the interview despite multiple follow-up attempts. The remaining 36 individuals of whom expressed interest did not respond after information sheets were supplied. We have added the above commentary to our results section.

Location: Pg. 14

In table 2: Years of experience – 1 to 5 or 2 to 5? I thought one year of experience was an exclusion criteria. Thank you very much for identifying this point of potential confusion - the table has been amended. Table 3.

It is interesting to be persuasive when writing the paper, however, I think it is too much. Consider taking off some words such as “important” or “illuminate”. 

Thank you for this suggestion - we have altered our phrasing in areas that may be construed as too persuasive.

Location: Pgs. 2, 34, 35

The themes are well presented and highlight several issues related to the low-resource environment challenges.

Discussion is well written

Thank you - we are grateful for your kind comments

Conclusion Focus on the leading suggestions derived from the study. It is the same as the abstract. Consider being more concise in the abstract. Finally, I’d like to congratulate the authors for their hard work and shed light on this issue.

Thank you for your comments. We have incorporated more of our leading suggestions within our Conclusions.

Location: Pg. 35

---

## [Decision Letter · Decision Letter 1]

7 Sep 2022

Neurotrauma clinicians’ perspectives on the contextual challenges associated with traumatic brain injury follow up in low-income and middle-income countries: a reflexive thematic analysis

PONE-D-22-03840R1

Dear Dr. Smith,

We’re pleased to inform you that your manuscript has been judged scientifically suitable for publication and will be formally accepted for publication once it meets all outstanding technical requirements.

Kind regards,

Hanna Landenmark

Staff Editor

PLOS ONE

Additional Editor Comments (optional):

Reviewers' comments:

Reviewer's Responses to Questions

**Comments to the Author**

1. If the authors have adequately addressed your comments raised in a previous round of review and you feel that this manuscript is now acceptable for publication, you may indicate that here to bypass the “Comments to the Author” section, enter your conflict of interest statement in the “Confidential to Editor” section, and submit your "Accept" recommendation.

Reviewer #1: All comments have been addressed

Reviewer #2: All comments have been addressed

Reviewer #3: (No Response)

2. Is the manuscript technically sound, and do the data support the conclusions?

Reviewer #1: Yes

Reviewer #2: Yes

Reviewer #3: (No Response)

3. Has the statistical analysis been performed appropriately and rigorously? 

Reviewer #1: N/A

Reviewer #2: N/A

Reviewer #3: (No Response)

4. Have the authors made all data underlying the findings in their manuscript fully available?

Reviewer #1: Yes

Reviewer #2: Yes

Reviewer #3: (No Response)

5. Is the manuscript presented in an intelligible fashion and written in standard English?

Reviewer #1: Yes

Reviewer #2: Yes

Reviewer #3: (No Response)

6. Review Comments to the Author

Reviewer #1: (No Response)

Reviewer #2: Congratulations, I believe mine and the others reviewers' comments have been addressed correctly. Thank you all for your efforts and hope this information will help improve neurotrauma follow-up around the world.

Reviewer #3: (No Response)

7. PLOS authors have the option to publish the peer review history of their article (what does this mean?). If published, this will include your full peer review and any attached files.

Reviewer #1: No

Reviewer #2: **Yes: **Sérgio Brasil

Reviewer #3: **Yes: **Robson Luis Oliveira de Amorim

---

## [Editor Report · Acceptance letter]

8 Sep 2022

PONE-D-22-03840R1 

Neurotrauma clinicians’ perspectives on the contextual challenges associated with traumatic brain injury follow up in low-income and middle-income countries: a reflexive thematic analysis 

Dear Dr. Smith:

I'm pleased to inform you that your manuscript has been deemed suitable for publication in PLOS ONE. Congratulations! Your manuscript is now with our production department. 

Kind regards, 

on behalf of

Dr. Hanna Landenmark 

Staff Editor

PLOS ONE